# Vaginal Bacteria in Mares and the Occurrence of Antimicrobial Resistance

**DOI:** 10.3390/microorganisms10112204

**Published:** 2022-11-08

**Authors:** Pongpreecha Malaluang, Elin Wilén, Sara Frosth, Johanna Lindahl, Ingrid Hansson, Jane M. Morrell

**Affiliations:** 1Department of Clinical Sciences, Swedish University of Agricultural Sciences (SLU), 75007 Uppsala, Sweden; 2Faculty of Veterinary Sciences, Mahasarakham University, Maha Sarakham 40000, Thailand; 3Evidensia Horse Clinic, 96174 Boden, Sweden; 4Biomedical Science and Veterinary Public Health, Swedish University of Agricultural Sciences (SLU), 75007 Uppsala, Sweden

**Keywords:** vaginal bacteria, mares, AMR

## Abstract

Antibiotics are added to semen extenders in insemination doses but their effect on the vaginal microbiota of the inseminated female is unknown. The objectives of this study were to define the equine vaginal microbiota and its antimicrobial resistance, and to determine whether it changes after exposure to antibiotics in semen extenders. Vaginal swabs were taken prior to sham-insemination (day 0), and again on days 3, 7, and 14 after insemination. Isolated bacteria were identified by MALDI-TOF and tested for antimicrobial susceptibility by microdilution. The bacteria isolated from the vagina differed according to reproductive status (brood mare or maiden mare), location (north or middle of Sweden), and the stage of the estrous cycle. Five bacterial species were frequently isolated from mares in both locations: *Escherichia coli*, *Staphylococcus capitis*, *Streptococcus equisimilis*, *Streptococcus thoraltensis*, and *Streptococcus zooepidemicus*. Overall, vaginal bacteria isolated from inseminated mares showed higher antibiotic resistance than from non-inseminated mares, suggesting a possible link between exposure to antibiotics in the semen extender and the appearance of antimicrobial resistance. The whole-genome sequencing of *E. coli* isolates from inseminated mares revealed some genes which are known to confer antimicrobial resistance; however, some instances of resistance in these isolates were not characteristic of induced AMR.

## 1. Introduction

Misuse and overuse of antimicrobials, especially in preventing infections or for growth promotion, are accelerating antimicrobial resistance (AMR), which has been described as one of the world’s biggest public health concerns [1]. The more prudent use of antibiotics has been advocated to minimize the development of resistance [2]. Thus, antibiotics should only be used for therapeutic purposes. However, this is not always the case as, for example, antimicrobials are added to insemination doses to prevent disease transmission and the deterioration of sperm quality during storage. Their use is specified by national and international requirements for the sale of semen obtained from production animals, including horses [3].

Previous studies concerning AMR in the equine reproductive tract primarily refer to animals with fertility problems and have been conducted in many countries since the 1980s, such as France, Germany, India, Italy, Slovakia, Sweden, Turkey, and the US, as reviewed by Malaluang et al. [4]. These studies cultured bacteria from uterine swabs, uterine lavage, vaginal swabs, and clitoral swabs. Resistance to antibiotics was detected in *Enterococcus* spp., *Enterobacter* spp., *Escherichia coli*, *Gardnerella vaginalis*, *Klebsiella* spp., *Micrococcus* spp., *Proteus* spp., *Pseudomonas* spp., *Staphylococcus intermedius*, *Streptococcus* spp., *Streptococcus equinus*, *Streptococcus equi* subsp. *Zooepidemicus*, and *Taylorella equigenitalis*, as reviewed recently [4]. The mares included in these studies were examined by the veterinarian because of fertility problems, and many had already been treated with various antibiotics over prolonged periods. Few studies have been performed regarding antimicrobial resistance in the vaginal flora of healthy mares without fertility issues, and the impact of different environments or the estrous cycle on vaginal flora has not been determined. Moreover, it is unknown whether the exposure of vaginal bacteria to antibiotics in semen extenders can alter antimicrobial resistance patterns and whether such changes are transient or long-lasting. Therefore, the objectives of the current study were (i) to describe the vaginal flora of two populations of mares in Sweden, using samples from brood mares and non-breeding mares; and (ii) to conduct a pilot study to determine suitable sampling times for a larger study on the possible changes in antimicrobial resistance after exposure to antibiotics in semen extenders.

## 2. Materials and Methods

### 2.1. Non-Inseminated Mares

Study 1 was a cross-sectional observational study: Swab samples were taken from the vagina at estrous in healthy mares. The study mares (n = 31) included both brood mares (n = 21) from a stud farm in Boden, in northern Sweden, which had not been inseminated since the last breeding season, as well as mares (n = 10) that had never been inseminated or had not been inseminated within the previous ten years (Figure 1a). These mares had unknown therapeutic profiles. In total, 6 of these last 10 mares were kept at the same stud farm as the 21 brood mares; 2 of these were new brood mares. The remaining four mares were housed at nearby farms. The horses were all housed under similar conditions and were of different breeds, including Swedish Standard Trotter, North-Swedish Trotter, and Arabian Thoroughbred. Their ages ranged from 2 to 20 years old, with a mean age of 9 ± 4 years. These mares were sampled once, as close as possible to the time of ovulation.

### 2.2. Inseminated Mares

Study 2 was a longitudinal experimental study: Swab samples were taken from the equine vagina following insemination with a semen extender (Figure 1b). The barren mares (n = 10) included in this study were housed at the Faculty for Veterinary Medicine and Animal Science, Swedish University of Agricultural Sciences, Uppsala. The inclusion criterion was that they had never been inseminated or had not been inseminated within the previous ten years (Figure 1b). These mares had not received antibiotics in the last six months. The mares were identified to be in standing heat from their behavior towards the stallion and were judged to be in the immediate pre-ovulatory phase based on ultrasound examination. They were sham-inseminated with a semen extender (INRA-96; IMV Technologies, L’Aigle, France). This extender contained the antimicrobials penicillin and gentamicin, as well as an antifungal, amphotericin B. Swabs of the vagina were taken before the semen extender was deposited in the uterus (day 0; D0), followed by sampling three (D3), seven (D7), and fourteen days later (D14). The D3 group included five mares which were swabbed on day five (D5) instead of D3. Samples taken at D0 are also included in the results from non-inseminated mares.

Ethical approval for swabbing was available prior to the study (number 5.8.18-15533/2018).

### 2.3. Sampling Technique

With the mares standing in examination stocks, the tail was wrapped before cleaning the perineal area and vulva with soap and lukewarm water. The vulva was dried with bleached paper and the procedure was repeated until the area was visibly clean. Sample contamination was minimized using a rectal glove, liquid paraffin, and a double-guarded occluded swab. The sampling site was approximately three centimeters distal to the vaginal fornix in the cranial vagina.

The swabs were transferred directly after sampling into Amie’s agar gel with charcoal (Copan Diagnostics, Inc., Murrieta, CA, USA). The swabs from inseminated mares were immediately transferred to the laboratory at SLU, where the analysis was initiated the same day. The samples from non-inseminated mares were stored at refrigerator temperature to prevent bacterial overgrowth until they were sent to the laboratory on Monday-Wednesday. The bacteriological analysis was initiated on the day of arrival at the lab.

### 2.4. Bacteriological Analysis

The swab samples were plated directly onto two blood agars (either bovine or horse) agars, lactose purple agar, MacConkey agar, Baird Parker agar, and De Man, Rogosa, and Sharpe (MRS) agar. One blood agar plate was incubated anaerobically at 37.0 ± 1 °C for 24 + 24 h, the other plates, except MRS-agar, were incubated aerobically at 37± 1 °C. Bacterial growth was recorded after 24 and 48 h incubation for all plates except the MRS-agar plates that were incubated anaerobically at 25 °C for five days before examination. Bacterial colonies with different macromorphology on the initial agar plates were noted and re-cultured on two blood agar plates (for aerobic and anaerobic culture conditions) and incubated for 24 to 48 h in a 37 °C incubator to obtain a pure culture. The isolates were identified at the species level by Matrix-Assisted Laser Desorption Ionization Time of Flight Mass Spectrometry (MALDI-TOF MS). The mass spectra of the bacterial isolates were compared automatically with those of known bacterial strains in the database (Bruker Daltonics, Billerica, MA, USA). All isolates were preserved in cryotubes with brain heart infusion (BHI) broth with 15% glycerol at −70 °C for subsequent antimicrobial susceptibility testing.

### 2.5. Antimicrobial Susceptibility Testing

Study 1: testing for antimicrobial resistance was performed on up to 22 selected isolates (mean 9) identified from all samples from the non-inseminated mares. In study 2, the testing was performed on 5 to 20 selected bacterial species (mean 10) identified from all sampling times (D0, D3, D7, and D14) from the same mare to compare antimicrobial resistance before and after sham-insemination. Susceptibility to selected antibiotic substances was assessed with *Thermo Scientific™ Sensititre ™ STAFSTR* (for *Streptococcus* spp. and *Staphylococcus* spp.), *Thermo Scientific™ Sensititre™ EUVENC* (for *Escherichia coli*), and *Thermo Scientific™ Sensititre™ EUVENSEC* (for *Enterococcus faecalis*) (Thermo Fisher Scientific, Waltham, MA, USA). Antibiotic minimum inhibitory concentration (MIC) was determined for *Escherichia coli*, *Enterococcus faecalis*, *Staphylococcus* spp., and *Streptococcus* spp. using broth microdilution following the standards of the Clinical and Laboratory Standards Institute [5]. Epidemiological cut-off (ECOFF) values for determining susceptibility were obtained from the European Committee on Antimicrobial Susceptibility Testing (EUCAST), (Breakpoint tables for interpretation of MICs, Version 11.0, 2021. http://www.eucast.org, accessed on 7 November 2022). The ECOFF values classify isolates with acquired reduced susceptibility as ‘non-wild type’. In this paper, non-wild type isolates are called ‘resistant’, in agreement with the Swedish Veterinary Antibiotic Resistance Monitoring report [6]. Multidrug resistance was defined as resistance to three or more antibiotic classes. For example, resistance to ciprofloxacin and nalidixic acid was considered resistance to one antibiotic class (quinolones).

### 2.6. Whole-Genome Sequencing

Since we could not be certain that isolates from any mare showing phenotypic resistance at any time during testing were of the same strain that showed phenotypic susceptibility at any other time, whole-genome sequencing (WGS) was performed. Only *E. coli* was subjected to WGS as the number of isolates of the other bacterial species was limited. All *E. coli* isolates from the individual mares with at least one isolate phenotypically resistant to any antibiotics at any time point were selected for WGS. The WGS was performed on 21 *E. coli* isolates in total, showing increased or decreased resistance to antibiotics according to the antimicrobial susceptibility testing. “Increased resistance” means that the proportion of bacteria from an individual mare showing resistance was higher later in the sampling sequence, whereas “decreased resistance “means that the proportion of bacteria showing resistance was lower later in the sampling sequence.

All isolates were re-cultured twice from single colonies on horse blood agar plates for 24 h at 37 °C in an aerobic atmosphere to ensure a pure culture, prior to DNA extraction using the EZ1 DNA Tissue Kit (Qiagen, Hilden, Germany) and the bacterial protocol on the EZ1 Advanced XL instrument (Qiagen) according to the manufacturer’s instructions. The DNA was eluted in a total volume of 100 µL, and the concentration was measured using the Qubit ds DNA High Sensitivity Assay Kit (Invitrogen, Carlsbad, CA, USA) on the Qubit^®^ 2.0 Fluorometer (Invitrogen). According to the manufacturer’s instructions, sequencing libraries were prepared using the Nextera XT DNA Library Preparation Kit (Illumina Inc., San Diego, CA, USA). The quality of the libraries was checked using the High Sensitivity DNA ScreenTape Analysis D1000 (Agilent Technologies, Inc., Santa Clara, CA, USA) on the 4150 TapeStation System (Agilent Technologies, Inc.). Quantification was done by the Qubit ds DNA High Sensitivity Assay Kit (Invitrogen) on the Qubit^®^ 2.0 Fluorometer (Invitrogen). The whole-genome sequencing was performed using the NextSeq 500/550 Mid Output kit V2.5 with 2 × 150-bp paired-end reads (Illumina Inc.) on a NextSeq 500 system (Illumina Inc.) at SLU.

Obtained sequences were analyzed using the Ridom SeqSphere + v7.0.5 software (Ridom GmbH, Münster, Germany). Genome assembly was performed de novo using SKESA (Souvorov et al., 2018) through a pipeline script in Ridom SeqSphere+ (Ridom GmbH). Core genome MLST (cgMLST) analysis was done using the E. *coli* cgMLST task template v1.0 in Ridom SeqSphere+ (Ridom, GmbH) containing 2513 loci. A minimum spanning tree (MST) based on cgMLST data was generated in Ridom SeqSphere+ (Ridom, GmbH) to investigate the relationship between isolates. Default parameters were used for generating the MST and missing alleles were ignored in the pairwise comparisons. A cluster distance threshold with a maximum of 10 cgMLST target differences was used to indicate the relationship, which was the default in the software. AMRFinderPlus [7] and ResFinder 4.1 [8,9,10] were used to detect the genes and point mutations associated with antimicrobial resistance. AMRFinderPlus was run via Ridom SeqSphere+ (Ridom, GmbH).

### 2.7. Statistical Analysis

Differences in the proportions of resistant bacteria between non-inseminated and inseminated mares were analyzed using the Chi-squared test, or Fisher’s Exact test if there were few observations for individual cells. A *p*-value of 0.05 or less was considered statistically significant.

## 3. Results

### 3.1. Bacterial Isolation

In study 1 (31 non-inseminated mares), 485 isolates of 28 different species of bacteria were detected. In total, 1 to 25 isolates (mean 15) were identified from the 31 non-inseminated mares.

In study 2 (10 sham-inseminated mares), 65 isolates were identified from 19 bacteria species. In total, 1 to 14 isolates (mean 4) were identified from mares on D0, 4 to 11 isolates (mean 7) were identified from mares on D3, 3 to 10 isolates (mean 5) were identified from mares on D7, and 2 to 9 isolates (mean 5) were identified from mares on D14.

A total of 41 different species of bacteria were detected. Five bacterial species were isolated from mares at both sites in Sweden, *Escherichia coli*, *Staphylococcus capitis*, *Streptococcus equisimilis*, *Streptococcus thoraltensis*, and *Streptococcus zooepidemicus*, with *E. coli* being dominant (40%). Barren and brood mares had different bacterial species; only two bacteria species, *Escherichia coli* and *Streptococcus zooepidemicus*, were common to all mares regardless of their reproductive status (Figure 2).

A distinct bacterial flora was found in specific sub-groups of mares (Table 1), with a greater diversity of bacteria being found in the mares in Boden. In addition, more species of Gram-negative bacteria were isolated from the mares in Boden than from the mares in Uppsala.

The numbers of isolates observed on D0, D3, D7, and D14 were 66, 89, 59, and 63, respectively, with a higher number of isolates being observed on D3 than at the other time point (*p* < 0.0001).

### 3.2. Antimicrobial Susceptibility

Resistance to at least one of the four antibiotics, chloramphenicol, colistin, sulfamethoxazole, and trimethoprim, was found for 24 *E. coli* isolates. There were significant differences between inseminated and non-inseminated mares, with *E. coli* from inseminated mares showing a higher resistance to trimethoprim (*p* = 0.00003) and chloramphenicol (*p* = 0.0309). In contrast, *E. coli* isolates from non-inseminated mares showed a higher resistance to sulfamethoxazole (*p* = 0.0273). However, the resistance of E. coli isolates to colistin was not significantly different between inseminated and non-inseminated mares. (Table 2).

Resistance to one to five of the eight tested antibiotics was found in 61 *Streptococcus* spp. isolates, of which, 2 isolates were multidrug resistant (resistant to at least three classes of antibiotics). In this case, no ECOFFs were given in the EUCAST for cefoxitin, enrofloxacin, fusidic acid, and gentamicin. There was a difference in resistance between inseminated and non-inseminated mares, with resistance to erythromycin being higher in non-inseminated mares (*p* = 0.013), whereas some *Streptococcus* spp. isolates were resistant to clindamycin, penicillin, tetracycline, and trimethoprim/sulfamethoxazole. However, no difference was found between inseminated and non-inseminated mares for resistance to clindamycin, penicillin, tetracycline, and trimethoprim/sulfamethoxazole (Table 3).

All *Staphylococcus* spp. isolates were susceptible to fusidic acid, tetracycline, and trimethoprim/sulfamethoxazole. There was no EUCAST cut-off MIC value for cefalotin, cefoxitin, clindamycin, enrofloxacin, and nitrofurantoin, meaning that an assessment of resistance could not be performed. Eight of twenty *Staphylococcus* spp. isolates were resistant to one to four of the seven tested antibiotics, including erythromycin, gentamicin, oxacillin, and penicillin; two *Staphylococcus* isolates from the inseminated mares were classified as multi-resistant (Table 4).

All *Enterococcus faecalis* isolates were susceptible to all six antibiotics tested, including ampicillin, ciprofloxacin, gentamicin, linezolid, teicoplanin, and tigecycline. There was no EUCAST cut-off MIC value for chloramphenicol, daptomycin, erythromycin, tetracycline, vancomycin, and quinupristin/dalfopristin, which means that an assessment of resistance could not be performed (Table 5).

The proportion of antimicrobial-resistant isolates of bacteria from inseminated and non-inseminated mares is shown in Table 6. In both groups of mares, most isolates were susceptible, or only a few isolates (below 50%) showed resistance, to the antibiotics tested. However, *S. equisimilis* and *S. zooepidemicus* from inseminated mares were often resistant to tetracycline, and 10 percent of *Staphylococcus* spp. from inseminated mares were resistant to oxacillin. In non-inseminated mares, resistance to sulfamethoxazole was often found among *E. coli* isolates and to erythromycin in *S. zooepidemicus*.

In inseminated mares, some bacterial species were shown to be resistant following exposure to antibiotics in the semen extender (Table 7). Thus, *E. coli* isolates were observed.

Thus, E. *coli* isolates were observed to be resistance to trimethoprim (32.4%), chloramphenicol (11.8%), and colistin (2.9%). *S. zooepidemicus* isolates showed resistance to tetracycline (23.8%), clindamycin (14.3%), and erythromycin (9.5%). *S. gallolyticus* isolates had resistance to erythromycin (75.0%), clindamycin (75.0%), tetracycline (75.0%), and trimethoprim/sulfamethoxazole (75.0%). *S. capitis* isolates were observed to be resistant to gentamicin (25.0%). *S. haemolyticus* isolates showed resistance to fusidic acid (14.3%) and gentamicin (14.3%). However, some bacteria were shown to be resistant to antibiotics both before and after exposure to the semen extender. *S. equisimilis* isolates were observed to be resistant to tetracycline (50.0% and 16.7% before and after exposure to the semen extender, respectively). One of the *Staphylococcus* spp., the isolate that was resistant to oxacillin, was sampled before and another three days after insemination. *S. haemolyticus* isolates were also resistant to penicillin before and after exposure (100.0% and 28.6%, respectively). Moreover, some bacteria were shown to be resistant only before exposure to antibiotics in the semen extender, e.g., in *S. capitis* resistance to penicillin (50.0%) was found, whereas, in *S. epidermidis*, resistance to penicillin (100.0%), fusidic acid (100.0%) and erythromycin (100.0%) was observed. It was not possible to detect changes in antimicrobial resistance because no identical bacteria were found in individual mares at all time points. Furthermore, it was not possible to determine visually if the same isolates were being tested. Therefore, the *E. coli* isolates were selected to be subjected to WSG.

### 3.3. Whole-Genome Sequencing

The purpose of the WGS was to determine whether *E. coli* isolates of different antimicrobial susceptibility sampled at different time points were likely to be of the same strain or not. Bacterial species other than *E. coli* were not subjected to WGS due to the limitation in the number of isolates detected at any time point.

Twenty-one *E. coli* isolates from inseminated mares at four time points, before (D0) and after (D3/D5, D7, and D14) insemination, could be differentiated into three clusters by cgMLST analysis (Figure 3). There was, at most, one allele’s difference between isolates within each cluster, and therefore isolates within each cluster were considered to be highly genetically related, i.e., from the same strain. Clusters 1 and 3 consisted of isolates from one mare each, whereas cluster 2 consisted of isolates from two different mares. The finding of genetically related isolates in two different mares indicates a possible spread within the herd.

Cluster 1 comprised ten isolates from mare A, of which all were phenotypically resistant to trimethoprim. In addition, two of the isolates were phenotypically resistant to chloramphenicol (P136_03 and P229_14); five belonged to cluster 2, of which three were phenotypically sensitive to all tested antibiotics (P45_03, P49_03, and P157_05), and two (P85_07 and P160_05) were phenotypically resistant to colistin and chloramphenicol, respectively. The colistin-resistant *E. coli* was isolated from mare B and the chloramphenicol-resistant isolate from mare C. The three isolates in cluster 3, P162_05, P232_14, and P237_14 (from mare C), were all phenotypically sensitive to all tested antibiotics.

### 3.4. AMR Genes

All *E. coli* isolates phenotypically resistant to trimethoprim had gene *dfrA14* which is responsible for trimethoprim resistance. Similarly, four chloramphenicol-resistant isolates had the gene *mdf(A)* which is responsible for chloramphenicol resistance. However, one isolate that was phenotypically resistant to colistin had none of the known colistin-resistance genes.

Ten *E. coli* isolates without phenotypic resistance had the sulfonamide resistance gene *sul2*, twenty-one isolates had the beta-lactam resistance gene *blaEC*, and ten isolates had *aph(6)-Id* gene, a streptomycin-resistance gene. In addition, the gene *mdf(A)* conferring resistance to broad-spectrum drugs via an efflux pump, was found in 19 isolates.

## 4. Discussion

### 4.1. Bacteria Isolation

The aim of this study was to describe the vaginal bacterial flora of mares in Sweden and to determine if antimicrobial resistance was present. A further aim was to determine suitable sampling times for a larger study in which the effects of transient exposure to antimicrobial substances in semen extenders on AMR would be investigated. A total of 530 bacteria were isolated from the vaginal swabs, with *E. coli* being isolated the most frequently (40.0%). This result is in agreement with earlier studies on vaginal bacteria in mares, where *E. coli* were the most commonly isolated bacteria in Korea (19.8%; [11]) and India (21.7%; [12]). There are, however, few previous studies on vaginal bacteria in mares; most studies focused on uterine bacteria [4]. The frequent occurrence of *E. coli* is not surprising since it is one of the most common intestinal microbes and one of the species with the shortest generation time.

Differences in bacterial species isolated among studies might be due to different environments and housing conditions for the horses and differences in culture conditions, such as types of agar plates, incubation atmosphere, and bacterial identification methods. In this study, the location of the mares tended to influence the vaginal flora, with a distinct bacterial flora being found in specific groups of mares. Of the 41 different bacterial species, only 5 were isolated from mares in both sites in Sweden: *E. coli*, *S. capitis*, *S. equisimilis*, *S. thoraltensis*, and S. *zooepidemicus*. The flora differed according to the location of the mare i.e., differed among stud farms.

### 4.2. Antimicrobial Susceptibility

Bacteria isolated from the equine vagina in this study were susceptible to most of the tested antibiotics with a range between 0% and 100%. This result is in contrast to most other published reports of bacteria from the equine reproductive tract, which cover uterine bacteria of mares with fertility issues, meaning that they might have received antibiotic treatment prior to sampling. Thus, the *Enterococcus faecalis* of non-inseminated mares in our study was susceptible to all 12 tested antibiotics, in contrast to previous studies in other countries where it was reported to be resistant to some antibiotics [12,13,14].

Some *E. coli* isolates of inseminated mares showed resistance to trimethoprim, chloramphenicol, and colistin, whereas there was no resistance to these antibiotics in *E. coli* isolated from non-inseminated mares. Age, housing condition, and whether or not the mare has been inseminated before might affect the susceptibility results since non-inseminated mares were mostly maiden mares, possibly with less chance of being in contact with antimicrobial agents than barren mares at the University. In this study, the resistance of *E. coli* isolates to trimethoprim and chloramphenicol is similar to previous reports [13,15,16,17]. However, colistin-resistance of *E. coli* isolates from the vagina and uterus of mares was not reported in previous studies.

*Staphylococcus* spp. isolates from inseminated mares were resistant to penicillin, oxacillin, erythromycin, and gentamicin in agreement with other studies [13,18,19].

*Streptococcus* spp. isolates of inseminated mares showed resistance to erythromycin, tetracycline, clindamycin, and trimethoprim-sulfamethoxazole. *Streptococcus equisimilis* isolates were resistant to tetracycline in both non-inseminated and inseminated mares. However, the small number of mares might confound the interpretation of this result. Moreover, differences in location, age, and previous therapeutic treatment might be involved. *Streptococcus equi* subsp. *zooepidemicus* isolates showed resistance to tetracycline, erythromycin, and clindamycin in both inseminated and non-inseminated mares. Trimethoprim/sulfamethoxazole resistance was observed in a non-inseminated mare. Different living conditions, e.g., grouping, age, previous treatment, personnel, or location, might contribute to the different resistance results observed. The resistance of *Streptococcus* spp. isolates were similar to previous reports [12,13,15,16,18,19,20].

Natural or acquired AMR arises from resistance genes. Many resistance genes occur in environmental bacteria that might potentially be transferred to pathogenic and non-pathogenic bacteria in the mare [21], or they might arise following exposure of the mares’ reproductive system to antibiotics. Low-level exposure to antimicrobial agents is considered to be one of the origins of AMR [22]. Such low-level exposure could arise because of exposure to antibiotics in semen extenders. 

In this study, inseminated mares were exposed to penicillin and gentamicin in the semen extender. Some antibiotic resistance was observed in non-inseminated mares, i.e., although they had not been exposed to antibiotics in semen extenders, they might have come into contact with other antibiotics through unknown therapeutic use, or they might have acquired resistance genes via gene transmission between vaginal and environmental bacteria. Thus, the variation in AMR apparently occurs among mares in groups with different backgrounds.

From our results, a higher number of isolates was found on D3 than at the other time points in the study. Therefore, this time point was chosen for a future larger study investigating changes in antibiotic resistance following exposure to antibiotics in semen extenders in future studies.

### 4.3. Whole-Genome Sequencing

Following antimicrobial resistance testing results, sequenced *E. coli* isolates from the inseminated mares could be differentiated into three MST clusters. Cluster 2 comprised a chloramphenicol-resistant isolate and a colistin-resistant isolate sampled at D5 and D7, respectively, in addition to three sensitive isolates (D3/D5), indicating the possible acquisition of AMR. A study revealed that low-level exposure to antibiotics could affect antimicrobial resistance [22]. In the case of chloramphenicol, the *mdf(A)* gene could explain this phenotypic resistance. However, since single isolates were selected for WGS based on microdilution results, it is not possible to be certain that the same bacterial strain was chosen on all sampling occasions. We cannot exclude that *E. coli* was present on D0; they could have been present in the mixed culture but were obscured by more dominant bacterial species.

One colistin phenotypically resistant isolate had none of the known responsible AMR genes, emphasizing that phenotypic resistance to these antibiotics may not be correlated to the presence of resistance genes. However, there is a measure of uncertainty in testing antimicrobial resistance by microdilution, as in most other analyses. In microdilution, one titer is considered within the margin of error of the analysis, and the *E. coli* isolate assessed as resistant in this study was only one titer away from being assessed as susceptible. An alternative explanation could be that the genes or the mechanism conferring resistance have not been detected yet or there may be a discrepancy between genotypic and phenotypic resistance.

## 5. Conclusions

Five bacterial species were isolated from all mares regardless of their reproductive status, *E. coli*, *S. capitis*, *S. equisimilis*, *S. thoraltensis*, and *S. zooepidemicus*. However, the vaginal bacterial flora differed according to the geographical location of the mares, although the most commonly isolated bacterium was *E. coli*, regardless of the location of the stud farm.

Overall, vaginal bacteria isolated from inseminated mares showed higher antibiotic resistance than non-inseminated mares, suggesting a possible link between the exposure to antibiotics in semen extender and the appearance of antimicrobial resistance. However, it is possible that other factors, such as increasing age, might increase the chance of exposure to antibiotics or increase the chance of non-genetic resistance conditions.

Whole-genome sequencing revealed some resistance of *E. coli* isolates from inseminated mares which are not characteristic of induced AMR. An example is that resistance to colistin was detected, although no AMR genes or mechanism responsible for colistin resistance was found, leading to the possibility of non-genetic resistance. Genetic resistance and persistence might play an important role in the resistance to trimethoprim and chloramphenicol. However, the number of isolates of each bacterial species from individual mares was limited, which might obscure some resistance. Further studies with a larger number of inseminated mares might reveal different results. Sampling 3 days after insemination appeared to be the most appropriate timing for detecting changes in resistance due to exposure to antibiotics in the semen extender. 

## Figures and Tables

**Figure 1 microorganisms-10-02204-f001:**
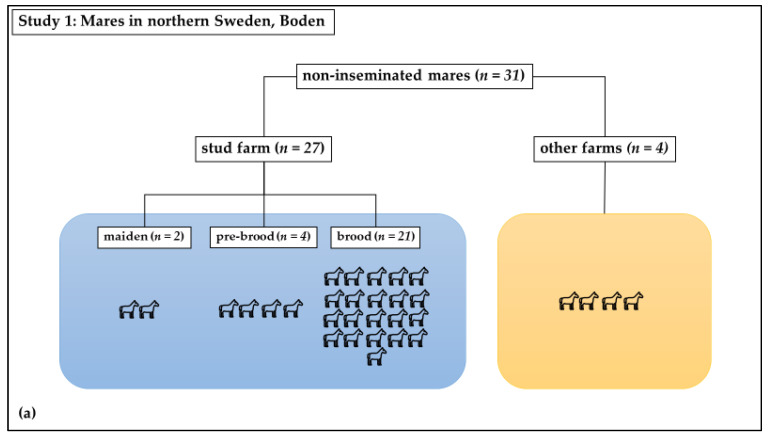
Distribution of mares in study groups: (**a**) Study 1: Non-inseminated mares were sampled once; (**b**) Study 2: Sham-inseminated mares were sampled on Day 0 (before insemination) and on days 3, 7, and 14 after insemination.

**Figure 2 microorganisms-10-02204-f002:**
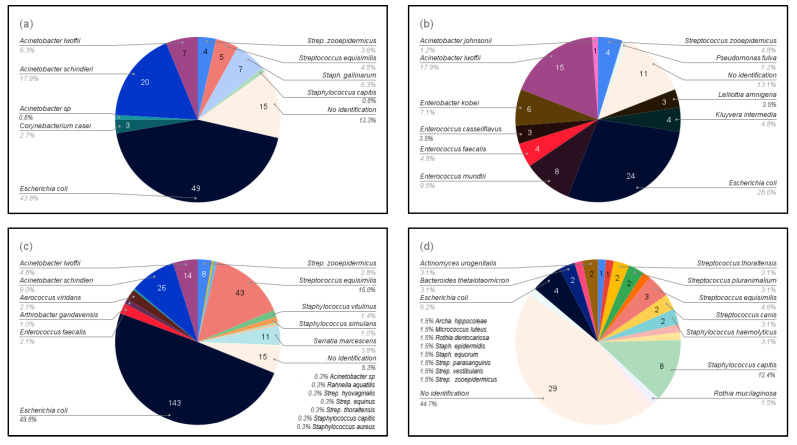
Bacteria isolated from the four groups of mares according to reproductive status: (**a**) Maiden mares in the stud farm in Boden (study 1); (**b**) Maiden mares in other farms in Boden (study 1); (**c**) Broodmares in the stud farm in Boden (study 1); and (**d**) Barren mares in Uppsala, including samples both before and after sham-insemination with semen extenders (study 2).

**Figure 3 microorganisms-10-02204-f003:**
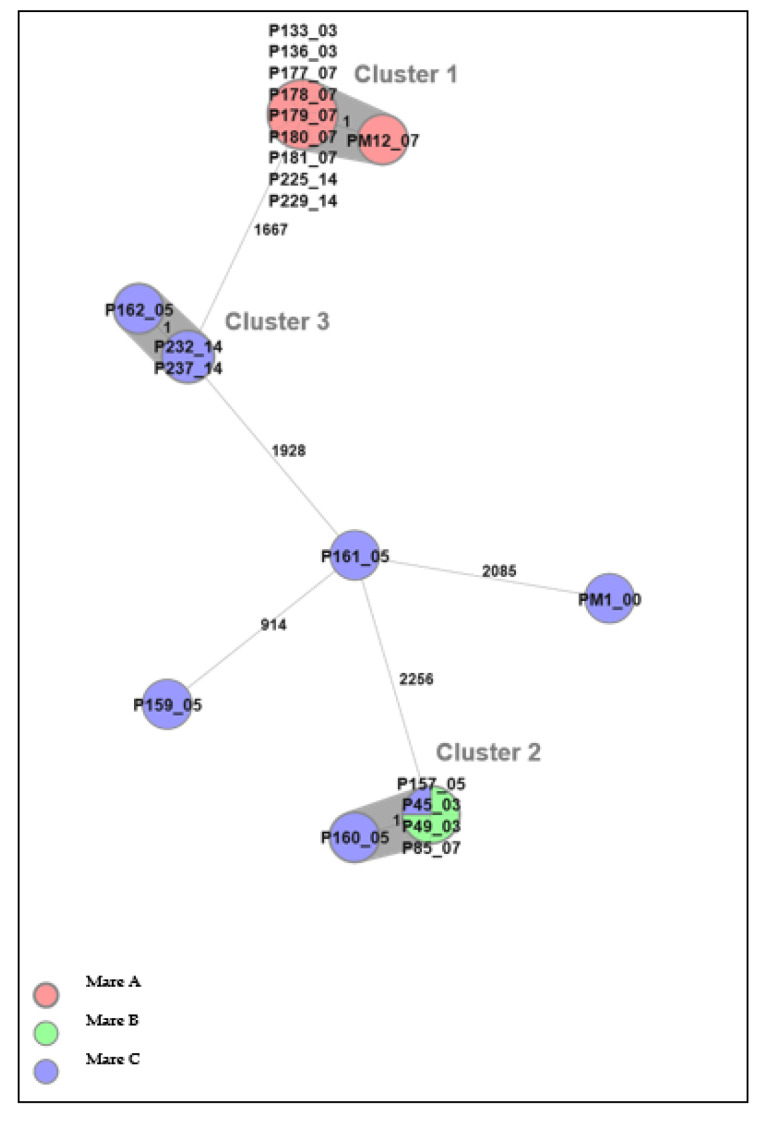
A minimum spanning tree (MST) was generated for the 21 *E. coli* isolates from the three mares in Uppsala, based on core genome multi-locus sequence typing (cgMLST) data. The text labels in circles denote the isolate ID followed by sampling day. Different colors denote individual mares. The grey background indicates identified clusters (genetically related isolates). Numbers next to the lines represent allelic differences, but the line length between isolates is not proportional to the numbers.

**Table 1 microorganisms-10-02204-t001:** Species of bacteria isolated from the vaginal flora of mares in two locations in Sweden, Boden and Uppsala.

Mares in Northern Sweden, Boden	
Gram-positive (n = 14)	*Corynebacterium casei*, *Enterococcus casseliflavus*, *Enterococcus faecalis*, *Enterococcus mundtii*, *Staphylococcus aureus*, *Staphylococcus capitis*, *Staphylococcus gallinarum*, *Staphylococcus simulans*, *Staphylococcus vitulinus*, *Streptococcus equinus*, *Streptococcus equisimilis*, *Streptococcus hyovaginalis*, *Streptococcus thoraltensis*, and *Streptococcus zooepidemicus*
Gram-negative (n = 13)	*Acinetobacter johnsonii*, *Acinetobacter lwoffii*, *Acinetobacter schindleri*, *Acinetobacter sp.*, *Aerococcus viridans*, *Arthrobacter gandavensis*, *Enterobacter kobei*, *Escherichia coli*, *Kluyvera intermedia*, *Lelliottia amnigena*, *Pseudomonas fulva*, *Rahnella aquatilis*, and *Serratia marcescens*
**Mares in middle Sweden, Uppsala**	
Gram-positive (n = 16)	*Actinomyces urogenitalis*, *Archanobacterium hippocoleae*, *Micrococcus luteus*, *Rothia dentocariosa*, *Rothia mucilaginosa*, *Staphylococcus capitis*, *Staphylococcus epidermidis*, *Staphylococcus equorum*, *Staphylococcus haemolyticus*, *Streptococcus canis*, *Streptococcus equisimilis*, *Streptococcus parasanguinis*, *Streptococcus pluranimalium*, *Streptococcus thoraltensis*, *Streptococcus vestibularis*, and *Streptococcus zooepidemicus*
Gram-negative (n = 2)	*Bacteroides thetaiotaomicron* and *Escherichia coli*

**Table 2 microorganisms-10-02204-t002:** Distribution of MICs (mg/L) and resistance of the 79 isolates of *Escherichia coli* from the cranial vagina of seven inseminated (I) and five non-inseminated (NI) mares. The results are shown as the percentage of isolates at different MIC values.

		Res	<0.015	0.03	0.06	0.12	0.25	0.5	1	2	4	8	16	32	64	128	256	512	>1024
Ampicillin	I (n = 34)	0								8.8	76.5	14.7							
	NI (n = 45)	0								15.6	71.1	13.3							
Azithromycin	I (n = 34)	0								2.9	44.1	53.0							
	NI (n = 45)	0								4.4	13.3	64.5	17.8						
Cefotaxime	I (n = 34)	0					100												
	NI (n = 45)	0					100												
Ceftazidime	I (n = 34)	0						100											
	NI (n = 45)	0						100											
Ciprofloxacin	I (n = 34)	0	85.3	14.7															
	NI (n = 45)	0	93.3	6.7															
Chloramphenicol *	I (n = 34)	11.8										88.2	11.8						
	NI (n = 45)	0										100							
Colistin	I (n = 34)	2.9							97.1		2.9								
	NI (n = 45)	0							100										
Gentamicin	I (n = 34)	0						58.8	41.2										
	NI (n = 45)	0						91.1	8.9										
Meropenem	I (n = 34)	0		100															
	NI (n = 45)	0		100															
Nalidixic acid	I (n = 34)	0									100								
	NI (n = 45)	0									97.8	2.2							
Sulfamethoxazole *	I (n = 34)	32.4										2.9	14.7	47.1	2.9				32.4
	NI (n = 45)	60.0											2.2			4.4	17.8	15.6	60.0
Tetracycline	I (n = 34)	0								70.6	11.8	17.6							
	NI (n = 45)	0								84.4	15.6								
Tigecycline	I (n = 34)	0					100												
	NI (n = 45)	0					100												
Trimethoprim *	I (n = 34)	32.4					2.9	55.9	8.8				2.9	29.5					
	NI (n = 45)	0					48.9	51.1											

White fields denote the range of dilutions tested for each antibiotic and bold vertical lines indicate the cut-off values used to define resistance. Grey fields denote the range outside of the dilutions tested. MICs equal to or lower than the lowest concentration tested are given as the lowest tested concentration. * Denotes a significant difference between inseminated and non-inseminated mares.

**Table 3 microorganisms-10-02204-t003:** Distribution of MICs (mg/L) and resistance of the 100 isolates of *Streptococcus* spp. from the cranial vagina of inseminated (I) and non-inseminated (NI) mares. The results are shown as the percentage of isolates at different MIC values. *Streptococcus* spp. from inseminated mares (n = 10), *Streptococcus equi* subsp. *zooepidemicus* from inseminated mares (n = 21) and non-inseminated mares (n = 13), and *Streptococcus equisimilis* from inseminated mares (n = 8) and non-inseminated mares (n = 48).

		Res	<0.03	0.06	0.12	0.25	0.5	1	2	4	8	16	32	>64
Cefalotin	I (n = 39)	0						100.0						
	NI (n = 61)	0						100.0						
Cefoxitin	I (n = 39)	-				5.1	7.6	71.8	10.3	2.6	2.6			
	NI (n = 61)	-				1.7	18.0	80.3						
Clindamycin	I (n = 39)	12.8					87.2	5.1	7.7					
	NI (n = 61)	3.2					96.8	1.6	1.6					
Enrofloxacin	I (n = 39)	-					23.1	76.9						
	NI (n = 61)	-					19.7	80.3						
Erythromycin *	I (n = 39)	12.8					87.2		12.8					
	NI (n = 61)	1.6					98.4	1.6						
Fusidic acid	I (n = 39)	-					53.8		46.2					
	NI (n = 61)	-					3.3		96.7					
Gentamicin	I (n = 39)	-							10.3	89.7				
	NI (n = 61)	-						19.7	8.2	72.1				
Nitrofurantoin	I (n = 39)	0										76.9	23.1	
	NI (n = 61)	0										85.2	14.8	
Oxacillin	I (n = 39)	0				100.0								
	NI (n = 61)	0				100.0								
Penicillin	I (n = 39)	5.2	87.2	7.6	5.2									
	NI (n = 61)	0	100.0											
Tetracycline	I (n = 39)	56.3				10.3	2.6	7.7	23.1	56.3				
	NI (n = 61)	77.0							23.0	77.0				
Trimethoprim/	I (n = 39)	7.7				84.6	7.7			7.7				
Sulfamethoxazole	NI (n = 61)	1.6				88.6	8.2	1.6		1.6				

White fields denote the range of dilutions tested for each antibiotic and bold vertical lines indicate the cut-off values used to define resistance. Grey fields denote the range outside of the dilutions tested. MICs equal to or lower than the lowest concentration tested are given as the lowest tested concentration. - = no cut-off values available. * Denotes a significant difference between inseminated and non-inseminated mares.

**Table 4 microorganisms-10-02204-t004:** Distribution of MICs (mg/L) and resistance of the 20 isolates of *Staphylococcus* spp. from the cranial vagina of inseminated mares (I). The results are shown as the percentage of isolates at different MIC values. (8 *Staphylococcus* capitis, 2 *Staphylococcus* epidermidis, 1 *Staphylococcus* equorum, 8 *Staphylococcus* haemolyticus, and 1 *Staphylococcus* warneri).

		Res	<0.03	0.06	0.12	0.25	0.5	1	2	4	8	16	32	>64
Cefalotin	I (n = 20)	-						95.0	5.0					
Cefoxitin	I (n = 20)	-				5.0	5.0	50.0	30.0	10.0				
Clindamycin	I (n = 20)	-					95.0		5.0					
Enrofloxacin	I (n = 20)	-				95.0	5.0							
Erythromycin	I (n = 20)	10					85.0	5.0		10.0				
Fusidic acid	I (n = 20)	10					90.0		10.0					
Gentamicin	I (n = 20)	10						90.0		10.0				
Nitrofurantoin	I (n = 20)	-										100.0		
Oxacillin	I (n = 20)	10				90.0		10.0						
Penicillin	I (n = 20)	35	55.0	5.0	5.0	5.0	15.0	15.0						
Tetracycline	I (n = 20)	0				45.0	35.0	10.0	10.0					
T/S	I (n = 20)	0				70.0	25.0		5.0					

T/S = Trimethoprim/Sulfamethoxazole (only Trimethoprim concentration). White fields denote the range of dilutions tested for each antibiotic and bold vertical lines indicate the cut-off values used to define resistance. Grey fields denote the range outside of the dilutions tested. MICs equal to or lower than the lowest concentration tested are given as the lowest tested concentration. - = no cut-off values available.

**Table 5 microorganisms-10-02204-t005:** Distribution of MICs (mg/L) and resistance of the four isolates of *Enterococcus faecalis* from the cranial vagina of non-inseminated mares (NI). The results are shown as the percentage of isolates at different MIC values.

		Res	<0.03	0.06	0.12	0.25	0.5	1	2	4	8	16	32	64	128	256	512	1024
Ampicillin	NI (n = 4)	0						100.0										
Chloramphenicol	NI (n = 4)	-									100.0							
Ciprofloxacin	NI (n = 4)	0						25.0	75.0									
Daptomycin	NI (n = 4)	-						75.0	25.0									
Erythromycin	NI (n = 4)	-						100.0										
Gentamicin	NI (n = 4)	0									100.0							
Linezolid	NI (n = 4)	0							50.0	50.0								
Q/D	NI (n = 4)	-										100.0						
Teicoplanin	NI (n = 4)	0					100.0											
Tetracycline	NI (n = 4)	-						75.0	25.0									
Tigecycline	NI (n = 4)	0		25.0	50.0	25.0												
Vancomycin	NI (n = 4)	-							100.0									

Q/D = Quinupristin/Dalfopristin. White fields denote the range of dilutions tested for each antibiotic and bold vertical lines indicate the cut-off values used to define resistance. Grey fields denote the range outside of the dilutions tested. MICs equal to or lower than the lowest concentration tested are given as the lowest tested concentration. - = no cut-off values available.

**Table 6 microorganisms-10-02204-t006:** Percentage of antimicrobial-resistant isolates of *Escherichia coli*, *Streptococcus* spp., *Streptococcus equi* subsp. *zooepidemicus*, *Streptococcus equisimilis*, *Staphylococcus* spp., and *Enterococcus faecalis* from the vaginas of healthy mares.

Antimicrobial	*E. coli*		*Streptococcus* spp.	*Streptococcus zooepidemicus*		*Streptococcus equisimilis*		*Staphylococcus* spp.	*E. faecalis*
	I (n = 34)	NI (n = 45)	I (n = 10)	I (n = 21)	NI (n = 8)	I (n = 8)	NI (n = 48)	I (n = 20)	NI (n = 4)
Trimethoprim	32	0	NA	NA	NA	NA	NA	NA	NA
Tetracycline	NA	NA	40	76	25	63	88	NA	NA
Chloramphenicol	12	0	NA	NA	NA	NA	NA	NA	NA
Colistin	3	0	NA	NA	NA	NA	NA	NA	NA
Gentamicin	0	0	NA	NA	NA	NA	NA	10	0
Sulfamethoxazole	32	60	NA	NA	NA	NA	NA	NA	NA
Penicillin	NA	NA	10	0	0	0	0	35	NA
Oxacillin	0	0	0	0	0	0	0	10	NA
Fusidic acid	NA	NA	NA	NA	NA	NA	NA	10	NA
Erythromycin	NA	NA	40	10	100	0	2	5	NA
Clindamycin	NA	NA	30	14	25	0	0	NA	NA
T/S	NA	NA	30	0	13	0	0	0	NA

T/S = Trimethoprim/Sulfamethoxazole, I = inseminated mare, NI = non-inseminated mare, NA = not applicable since the isolate was not tested for the antimicrobial substance or there were no ECOFFs given.

**Table 7 microorganisms-10-02204-t007:** Antimicrobial resistance of bacteria isolates from the vagina of 10 mares before and after insemination.

Bacteria	Day 0		Day 3		Day 7		Day 14	
	Isolate ID	Resistance	Isolate ID	Resistance	Isolate ID	Resistance	Isolate ID	Resistance
*E. coli*			P136	Tri and Chl	P177, P178, P179, P180, P181, P192, and PM12	Tri	P229	Tri and Chl
			P159 and P160	Chl	P85	Col	P225	Tri
			P133	Tri				
*S. equisimilis*	P5	Tet	P57	Tet				
*S. zooepidemicus*			P158	Tet			P148	Ery, Cli, and Tet
							P126, P151, and PM5	Tet
*Streptococcus* spp.	P11	Ery and Tet	P109	Ery, Cli, Tet, and T/S			P217 and PM6	Ery, Cli, Tet, and T/S
*Staphylococcus* spp.	P21	Pen, Oxa, Fus, and Ery	P198	Fus and Gen	P207	Pen	P245	Pen
	P2, P143, and P146	Pen	P156	Gen				
	P10	Ery	P76	Pen and Oxa				

Tri = trimethoprim, Chl = chloramphenicol, Col = colistin, Tet = tetracycline, Pen = penicillin, Ery = erythromycin, Cli = clindamycin, T/S = trimetoprim/sulfamethoxazol, Oxa = oxacillin, Fus = fusidic acid, and Gen = gentamicin.

## Data Availability

All data are supplied in the manuscript.

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
