# Peer review of "Vaginal Bacteria in Mares and the Occurrence of Antimicrobial Resistance"

_microorganisms, 2022, doi:10.3390/microorganisms10112204_

Round 1
Reviewer 1 Report
Line 34-35: Please re-phrase the sentence "antibiotics should be used for therapeutic purposes"
Line 55-56: Please be specific about objective 1 for antimicrobial resistance bacteria
Line 62: Please state the study design. How did you determine the sample size?
Line 74: Please state the study design. How did you determine the sample size?
Line 183: Please correct the sentence to "differences in the proportion of antimicrobial resistance of bacteria..................................
Statistical analysis: How did you determine objective 2 (and discussion line 394-398 & conclusion line 432-434)
--------------------------------------------------------------
Author Response
"Please see the attachment."

Reviewer 2 Report
The manuscript sounds good and is well-written.
Just have two minor comments.
Figure 1 should be improved. Correct topography errors and improve the visualization.
You mentioned the emergence of colistin-resistant mcr(-) isolates. Did you look at mutations in the LPS genes? Try to identify putative genes associated with this resistance. If not, try to recheck the MIC of your isolates to colistin. As I understood, the current MIC = 4 mg/l; thus, the MIC could differ by one dilution if we retest the isolates, and then they become sensitive (error of uncertainty).
Author Response
"Please see the attachment."
